# Germination Requirement and Suitable Storage Method of *Hydrocharis dubia* Seeds

**DOI:** 10.3390/biology13040246

**Published:** 2024-04-07

**Authors:** Suting Zhao, Hongsheng Jiang, Yang Liu, Ling Xian, Wenlong Fu, Saibo Yuan, Liyan Yin, Wei Li

**Affiliations:** 1Hubei Key Laboratory of Big Data in Science and Technology, Wuhan Library, Chinese Academy of Sciences, Wuhan 430071, China; zhaost@mail.whlib.ac.cn; 2Wuhan Botanical Garden, Chinese Academy of Sciences, Wuhan 430074, China; jhs@wbgcas.cn (H.J.); liuyang1025future@163.com (Y.L.); xianling@wbgcas.cn (L.X.); 3Hubei Key Laboratory of Wetland Evolution & Ecological Restoration, Wuhan Botanical Garden, Chinese Academy of Sciences, Wuhan 430074, China; 4Hubei Provincial Academy of Eco-Environmental Sciences, Wuhan 430070, China; fwl0415@126.com; 5Ecological Environment Monitoring and Scientific Research Center, Ecology and Environment Supervision and Administration Bureau of Yangtze Valley, Ministry of Ecology and Environment of the People’s Republic of China, Wuhan 430014, China; yuansaibo_ihb2013@163.com; 6School of Life and Health Sciences, Hainan University, Haikou 570228, China; 7School of Ecology and Environment, Tibet University, Lhasa 850000, China

**Keywords:** seed, germination, storage, *Hydrocharis dubia*, conservation

## Abstract

**Simple Summary:**

Understanding seed germination requirements and storage methods is important for successful conservation and restoration of aquatic vegetation. Main research issues are Hydrocharis dubia seed germination requirements and appropriate seed storage methods. It was found that high seed clustering density and light had positive effects on seed germination, while burial had negative effects on seed germination. Oxygen, water level and substrate had no significant effect on seed germination. Seed germination, water content and respiration rate were significantly affected by storage method. Seed germination was highest under the Ambient Water Temperature-Wet storage condition, followed by 4 °C-Wet and then 4 °C-Dry. Seeds did not germinate under the storage conditions of Ambient air temperature-Wet and Ambient air temperature-Dry.

**Abstract:**

Understanding of seed germination requirements and storage methods is very important to successfully conserve and restore aquatic vegetation. The main question addressed by the research was germination requirements and suitable seed storage methods of *Hydrocharis dubia* seeds. Furthermore, the water content and respiration rate of *H. dubia* seeds were studied under different storage conditions. The study found that light and high seed clustering density had a positive effect on germination, while burial had a negative effect. Germination percentages were 60.67 ± 6.11% and 28.40 ± 6.79% in light and dark conditions, respectively. Under clustering densities of 1 and 50, germination percentages were 6.00 ± 2.00% and 59.33 ± 0.67%, respectively. Germination percentages were 50.40 ± 5.00%, 3.20 ± 3.20%, and 0.80 ± 0.80% at depths of 0, 2, and 3 cm, respectively. Oxygen, water level, and substratum had no significant effect on seed germination. Storage method had a significant effect on seed germination, moisture content, and respiration rate. The germination percentages were 64.00 ± 1.67%, 85.20 ± 5.04%, and 92.80 ± 4.27% under the storage conditions of 4 °C-Dry, 4 °C-Wet, and Ambient water temperature-Wet for 2 years, respectively. The seeds had no germination under the storage conditions of Ambient air temperature-Wet and Ambient air temperature-Dry. Overall, the study indicates that seed germination of *H. dubia* is restricted by light, burial depth, and seed clustering density. Additionally, it was found that *H. dubia* seeds can be stored in wet environmental conditions at ambient water temperature, similar to seed banks. Specifically, the seeds can be stored in sand and submerged underwater at ambient water temperatures ranging from 4 °C to 25 °C. This study will help with the conservation and restoration of aquatic plants, such as *H. dubia*.

## 1. Introduction

Many aquatic plants have been threatened worldwide by adverse changes over the last few decades [1]. As primary producers, aquatic plants play a very important role in water ecosystems. Not only do they provide food and habitat for many other aquatic organisms, they also improve water quality by reducing nutrient availability [2]. Accordingly, the protection and restoration of aquatic plants is imperative. Seeds of aquatic plants are necessary for the restoration of the declined population, and re-seeding is an effective method to restore the disappeared vegetation [3]. Furthermore, regeneration from seeds is important for the diversity and composition of plant communities [4]. Consequently, understanding of seed storage methods and germination requirements is very important to conserve them successfully and to restore vegetation.

Many studies have indicated that temperature, substratum, oxygen concentration, light availability, burial depth, water depth, and seed clustering density may be important limiting factors for seed germination of aquatic macrophytes [5,6,7,8,9]. Numerous studies have demonstrated that light is necessary for high germination rates in most freshwater species [10,11,12]. In eutrophic lakes, oxygen level in sediment is very low, and oxygen is an important factor for seed germination of aquatic macrophytes [7,13]. Anaerobic conditions proved to accelerate seed germination of *Vallisneria natans* [14] and had no significant effect on seed germination of *Ottelia alismoides* [15]. Burial inhibits seed germination of many aquatic plants [6,9]. Sediment type often influences seed germination rate in aquatic plant seed germination [9,16]. Water level is also an important trigger for seed germination of some aquatic plants [13,17]. Density-dependent clustering effects on seed germination both increase and decrease germination when seeds are densely clustered [18], and high seed clustering density can enhance the germination of *O. alismoides* [19]. These studies indicate that light, oxygen availability, water level, burial depth, substrate, and clustering density can be important determinants of the seed germination of aquatic plants.

An effective seed storage method of aquatic plants is still under investigation. Previous studies have indicated that temperature, relative humidity, and oxygen of the storage conditions should be considered for seed storage [6]. Cold storage conditions led to a higher germination percentage for numerous aquatic species [20]. At low temperature (4 °C) storage conditions, dry storage conditions are more favorable than wet storage conditions for seeds of *Carex nebrascensis*, *O. acuminata*, *V. natans*, *Potamogeton wrightii*, *Stuckenia pectinata*, *Myriophyllum balladoniense*, and *Triglochin linearis*, while seeds of *Juncus balticus*, *J. ensifolius*, *J. tenuis*, *Alisma gramineum*, *O. alismoides*, and *P. lucens* had higher seed vigor after wet storage [6,21]. For seeds of *Zostera marina*, there was a negative relationship between seed germination and oxygen concentration [22]. Anaerobic storage condition was an effective storage condition for three species (*O. acuminata*, *O. alismoides* and *V. natans*) of Hydrocharitaceae [6].

The *Hydrocharis* genus acts as edificators and subedificators in freshwater vegetation communities. Furthermore, it is considered an excellent bioremediator due to its ability to bioaccumulate heavy metals on its roots [23]. Research on the genus has primarily focused on plant cytobiology [24], classification [25], and phylogeny [26,27,28]. Studies have also been conducted on the reproductive [29,30] and seed morphology [31,32] of the genus. Previous studies have shown that the ripe fruit of *H. laevigata*, one species of *Hydrocharis* genus, can release a mucilaginous mass of approximately 100 seeds. For aquatic plants, sexual reproduction might be a luxury investment in most cases, responsible for population restoration from extreme events and primarily working on an evolutionary time scale. However, research on *Hydrocharis* genus seed germination and storage is insufficient.

As a floating-leaved aquatic plant, *H. dubia* could be used as food for animals, as a vegetable, as green manure, or for medicine. It has medicinal properties for alleviating heat and dampness. *H. dubia* can reproduce through clonal growth and seed germination. The seeds can be easily collected. This makes seed banking conservation an effective way to protect them and re-seeding a particularly attractive method for restoration. However, there is little information about the factors influencing seed germination and the effective storage conditions for maintaining seed vigor. Understanding of *H. dubia* germination requirements and seed storage method are vital for its reproduction and conservation. The hypotheses of this research were as follows: (1) light, oxygen, water level, burial depth, substratum, and seed clustering density are important limiting factors for seed germination of *H. dubia*, and (2) the most appropriate method of storage for *H. dubia* seeds is one that is similar to the storage conditions in the seed bank. This study extends previous research on seed germination and seed storage of aquatic plants by filling in gaps in knowledge regarding the environmental conditions required for seed germination and the suitable storage conditions of floating-leaved plants like *H. dubia*.

## 2. Materials and Methods

### 2.1. Materials

The germination cycle of the *H. dubia* plant is illustrated with a timeline in Figure 1. The fruiting period of *H. dubia* occurs between August and October in China; it has high seed productivity and can produce a large amount of seeds every year. The fruit of *H. dubia* has six incompletely formed locules with superficial placentation [33] and contains about 200–400 seeds. The number of seeds in the fruit was measured as follows: in October 2014, ten *H. dubia* fruits were randomly collected from the Wuhan Botanical Garden and placed in ten mesh bags (10 cm in length and 8 cm in width) with a hole size of 0.5 mm, respectively. The mesh bags were placed inside a plastic bucket (25 cm diameter and 20 cm height) that was filled with water. The number of seeds in each fruit was then counted after they were dispersed. The seeds are very small (approximately 1.4 mm long and 0.6 mm wide).

### 2.2. Seed Collection

Mature fruits of *H. dubia* were collected from October to November 2013 and 2014 from Wuhan Botanical Garden, Chinese Academy of Sciences (30°32′ N, 114°25′ E). The seeds were removed from the fruits and washed with distilled water. Mature seeds (with brown testa color) were used in this study.

### 2.3. Germination Experiments

The germination experiments were conducted in April 2015 using the seeds collected in 2014. We utilized the single-factor design to study the effects of light availability, oxygen availability, water level, burial depth, substratum types, and clustering density on seed germination of *H. dubia* (Figure 2). All the germination experiments were conducted at 25 °C using a single factor design with 5 replicates and 50 seeds per replicate. Germination was considered to have occurred when the radical emerged from the seed coat. The number of germinated seeds was recorded daily, and the test was concluded when no new germination was observed for seven consecutive days.

#### 2.3.1. Effect of Light Availability on Germination

Seeds were placed in water using plastic culture dishes (diameter: 1 cm, height: 1 cm). The experiment was conducted under conditions where the oxygen concentration was in equilibrium between water and the atmosphere. Distilled water was used as the substratum. Dark conditions (control) were created by wrapping dishes with a double layer of aluminum foil. The control dishes were kept in complete darkness, while the light treatment dishes were exposed to a 12 h photoperiod with a photon flux density of 20 ± 5 μmol m^−2^ s^−1^ provided by warm white fluorescent lamps. Germination under dark conditions was checked using a safe lamp in the dark room.

#### 2.3.2. Effect of Oxygen Availability on Germination

Glass tubes (diameter: 58 cm, height: 38 cm) were used in this experiment. A total of 20 mL of distilled water was used as the substratum. To make the low-oxygen treatment, 20 mL of distilled water flushed with N_2_ gas for 15 min was added in the tubes, and the tubes were bubbled for 2 min with 800 mL min^−1^ of N_2_ gas before being sealed with rubber stoppers. The control tubes were opened for ventilation, and distilled water was added every other day to maintain the water level. The experiment was conducted under a 12 h photoperiod. The seeds were checked for germination through the transparent wall of the vials.

#### 2.3.3. Effect of Water Level on Germination

Seeds were incubated in cylindrical glass tubes (25 cm diameter and 40 cm height) at seven different water levels (0, 1, 5, 10, 20, 30, and 40 cm). Water levels of 0 cm were the control group. For the 0 cm water level, two layers of filter paper moistened with distilled water were placed in the bottom of the tube. The experiment was conducted under a 12 h photoperiod with oxygen concentration maintained in equilibrium between water and the atmosphere. Distilled water was added every day to maintain the designed water levels.

#### 2.3.4. Effect of Burial Depth on Germination

Five burial depths (0, 1, 2, 3, and 4 cm) were designed using cylindrical plastic containers (5 cm diameter and 8 cm height). The control group had burial depths of 0 cm. The experiment used pond sediment that had been sterilized at 120 °C for 30 min and was conducted under a 12 h photoperiod. Distilled water was added every day to maintain the water level at 1 cm above the substratum surface.

#### 2.3.5. Effect of Substratum on Germination

Four germination substrata (water, well-washed sand from the Yangtze River, mud from a pond in Wuhan Botanical Garden, and a mixture of sand and mud in equal volume proportions) were designed in the substratum experiment. The control group was the water substratum. The experiment was carried out using a plastic board with 48 holes, with 0.5 cm height substratum and 0.5 cm height distilled water in each hole. The experiment was conducted under a 12 h photoperiod, and the seeds were placed on the substrate surface.

#### 2.3.6. Effect of Clustering Density on Germination

The experiment was conducted using glass petri dishes with a diameter of 9 cm. Seeds were carefully arranged in 2 sets of cluster densities with 5 replicates: (a) all 50 seeds were mixed together (control); (b) seeds were separated from each other (50 seeds per dish). Seeds were placed on filter paper moistened with distilled water in petri dishes. The experiment was conducted with a 12 h photoperiod, and the oxygen concentration was maintained in equilibrium between the water and the atmosphere.

### 2.4. Storage Experiments

The seeds collected in 2013 were used in the storage experiment. The detail treatments of storage conditions were referenced by [6]. The storage treatments comprised a factorial combination of two environments and two temperatures (Figure 3, Table 1). The two storage environments comprised (i) a paper bag (dry) and (ii) water that was changed weekly (wet). The two storage temperatures comprised (i) low temperature (4 °C) to simulate the winter water temperature, achieved by using a refrigerator, and (ii) ambient air temperature to simulate the changes in a full year’s air temperature (ambient) (Figure A1). In addition, the control treatment was used to simulate natural conditions in a lake with the ambient water temperature ranging from 4 °C to 25 °C. We proposed a storage method for aquatic plants that is similar to a seed bank. This is an innovative approach to storing aquatic plants. Specifically, the seeds were placed in mesh bags that measured 10 cm in length and 8 cm in width. The hole size of the mesh bags was 0.5 mm. Mesh bags were placed at the bottom of plastic buckets measuring 25 cm in diameter and 20 cm in height, which were then filled with sand. Plastic buckets were submerged in a pool of water that measured 60 cm in length, 40 cm in width, and 80 cm in depth. Relative humidity was measured using the hygrometer (Deli 9013).

In order to explore the optimum storage method, seeds collected in 2013 were used to test germination in September 2015. A total of 50 seeds (5 replicates) in each storage condition were placed in plastic culture dishes (diameter: 1 cm, height: 1 cm). In order to keep the water level at 1 cm, distilled water was added every day. The germination temperature was 25 °C and the light period was 12 h. The photo irradiance was about 25 μmol m^−2^ s^−1^ provided by warm white fluorescent lamps (PAR; Li-Cor underwater sensor connected to a Li-Cor LI-1400 data logger) (Li-COR Inc., Lincoln, NE, USA). The oxygen concentration was maintained in equilibrium between the water and the atmosphere.

The seed moisture content was determined according to Zhao et al. [34]. In brief, 0.05 g seeds (with five replicates) were placed in a glass culture dish and weighed together. Then they were oven-dried at 125 °C for 2 h. After cooling for 24 h in a desiccator, seeds were weighed several times until a constant final seed weight was determined. The seed moisture content was calculated according to the percentage of fresh weight.

Seed respiration rate was determined at 25 °C using the O_2_ micro-electrode (Unisense, Aarhus, Denmark). The respiratory rate was corrected for oxygen consumption via the electrode, and the results were expressed as μmol O_2_ h^−1^ mg^−1^ FW.

## 3. Data Analysis

Data were analyzed using SPSS 22.0 software. The data were arcsine- or square-root transformed prior to analysis when necessary. The *t*-test was used to analyze the light, oxygen, or clustering effects. A one-way ANOVA was used to analyze the effects of water level, buried depth, and substrate on seed germination, as well as the impact of storage conditions on the germination, water content, and respiration of *H. dubia* seeds. Multiple comparisons were made using Tukey’s test. The confidence limits were set at 95%.

## 4. Results

### 4.1. Germination Requirement in Seeds of H. dubia

Light had a significant effect on the seed germination of *H. dubia* (Figure 4A, Table 2). The germination rate was 60.67 ± 6.11% under light, which was significantly higher than that under dark (28.40 ± 6.79%).

Oxygen availability had no significant effects on the seed germination of *H. dubia* (Figure 4B, Table 2). The results show that the germination rate was 69.50 ± 6.18% and 76.5 ± 3.59% under high and low oxygen conditions, respectively.

Water level had no significant effects on the seed germination of *H. dubia* (Figure 4C, Table 2). The germination rate was 47.33 ± 4.67%, 58.67 ± 2.91%, 54.67 ± 4.37%, 51.33 ± 1.76%, 58.00 ± 4.16%, 47.33 ± 4.81%, and 52.00 ± 0.00% under underwater conditions at depths of 0, 1, 5, 10, 20, 30, and 40 cm, respectively.

Burial depth had a significant effect on seed germination (Figure 4D, Table 2). Germination percentage for buried seeds was significantly lower than that for non-buried ones. The germination percentage was 50.4% at a 0 cm burial depth but significantly decreased to 32% and 0.8% at depths of 1 cm and 2 cm. Once the burial depth exceeded 3 cm, seeds failed to germinate.

The natural substrata had no significant effects on the seed germination of *H. dubia* (Figure 4E, Table 2). The germination rate was 47.20 ± 1.02%, 47.20 ± 1.02%, 50.40 ± 4.96%, and 54.67 ± 4.37% under substrata conditions of water, sand, mud, and a mixture of sand and mud in equal volume proportions, respectively.

Clustering density had a significant effect on the seed germination of *H. dubia* (Figure 4F, Table 2). Higher seed clustering density strongly increased the seed germination of *H. dubia*. The percentage of germination in the condition where the seeds were scattered in the dish was significantly lower than the seed germination percentage in the condition where 50 seeds were clustered together. The germination percentage was only 6.0% when the seeds were dispersed in the dish, while it was 59.3% at the density of 50 seeds per cluster.

### 4.2. Suitable Storage Method in Seeds of H. dubia

The germination percentages were 64.00 ± 1.67%, 85.20 ± 5.04%, and 92.80 ± 4.27% under the storage conditions of 4 °C—Dry, 4 °C—Water, and Ambient water temperature—Wet, respectively (Table 3). The seeds in the 4 °C storage environment (4 °C—Wet and 4 °C—Dry) had lower germination than that in Ambient water temperature—Wet storage treatments (Table 4, Figure A2). In the 4 °C storage environment, the germination in the wet storage condition was significantly higher than that in the dry storage condition. The seeds did not germinate under the Ambient air temperature—Dry storage environment. Seeds in the Ambient air temperature—Wet storage environment rotted before the germination experiment.

The water content of the seeds was 13.80 ± 0.62%, 35.44 ± 0.50%, 11.51 ± 0.57%, and 32.49 ± 1.13% under the storage conditions of 4 °C—Dry, 4 °C—Wet, Ambient air temperature—Dry and Ambient water temperature—Wet, respectively. Seed moisture content in dry storage treatments (4 °C—Dry and Ambient air temperature—Dry) was significantly lower than that in Ambient water temperature—Wet storage treatments. There was no significant difference in seed moisture between the two different conditions of 4 °C—Wet and Ambient water temperature—Wet.

The respiration rate of the seeds was 13.55 ± 1.36, 11.09 ± 0.51, 7.36 ± 2.25, and 15.25 ± 2.10 μmol O_2_ h^−1^ mg^−1^ FW under the storage conditions of 4 °C—Dry, 4 °C—Wet, Ambient air temperature—Dry, and Ambient water temperature—Wet, respectively. Seed respiration rate in Ambient air temperature—Dry storage treatments was significantly lower than that in Ambient water temperature—Wet.

## 5. Discussion

Light is the greatest ecological factor influencing the emergence of the seed bank. Our results show that light promoted the seed germination of *H. dubia*. Similar results were reported for seeds of *O. alismoides* [15,19]. Seeds of *H. dubia* are relatively small. Very small seeds tend to have an absolute germination requirement for light [35,36]. Light is a crucial environmental cue determining seed germination in some species. The red (R) and far-red light photoreceptor phytochrome regulates GA biosynthesis in germinating lettuce and Arabidopsis seeds. This effect of light is, at least in part, targeted to mRNA abundance of GA 3-oxidase, which catalyzes the final biosynthetic step to produce bioactive gas [37]. Decrease in transparency is often caused by phytoplankton in eutrophic waters [38]. Thus, the reduced benthic light not penetrating far into the sediment might be a limiting factor for seed germination of *H. dubia*.

Oxygen is required for the seed germination of many aquatic species [39]. Our results show that oxygen availability had no significant effects on *H. dubia* seed germination. This is consistent with the results from studies on seeds of *O. alismoides* [15], *V. natans* [14], *Z. marina* [22], and *Z. capricorni* [40]. Thus, sediment anoxia is not a limiting factor for *H. dubia* seed germination and population recruitment [15]. Eutrophic lake sediment is usually under the anoxia condition [41]. The mud at the bottom of eutrophic waters tends to become anoxic due to the accumulation of organic matter. The decomposition of organic matter in sediment consumes a significant amount of oxygen, leading to sediment anoxia [42]. Seed germination could be inhibited in such a environment. Seeds of *Trapella sinensis* have an almost absolute requirement for aerobic conditions to germinate [43]. Under low oxygen conditions, germination of *H. dubia* was not significantly inhibited. This may also be one of the reasons why *H. dubia* can live in eutrophic waters.

The seed germination requirements of submerged species in the water level condition are not fully understood [44,45]. Our results show that water level had no significant effects on seed germination of *H. dubia*. Similar results were reported for seeds of *M. spicatum* [46], *P. natans*, and *P. perfoliatus* [47]. Water level may influence the seed germination by affecting levels of available light and oxygen [46,48]. The submerged treatment significantly enhanced the seed germination of *P. malaianus* [46] while inhibiting the germination of *Hottonia palustris* [49] and *H. inflate* [50]. Typically, water level may influence levels of available oxygen and light, leading to different germination and dormancy responses of seeds. For seeds of *H. dubia*, light, not oxygen availability, was the limited factor. This demonstrated that the effect of water level on *H. dubia* seed germination had no relationship with oxygen availability, and seeds could obtain enough light to germinate even when they were submerged at a 40 cm water level. In this study, the highest water level was designed to be only 40 cm due to limitations in the experimental facility. This may not be sufficient to produce a substantial difference in available light.

Our results show that burial greatly inhibited the seed germination of *H. dubia*. These results are consistent with the results for seeds of *V. natans* [14], *O. alismoides* [15], *M. spicatum* [51], *M. spicatum*, and *P. malaianus* [46]. Light was the limited factor for seed germination of *H. dubia*. Studies on *O. alismoides* seeds have shown that stirring up the sediment slightly to bring seeds to the surface is beneficial to seed germination [15]. The limiting factors for seed germination of *H. dubia* are nearly the same with *O. alismoides*. This slight stirring up sediment method might also be used for the restoration of *H. dubia*. The depth at which the seeds of *H. dubia* are buried is a limiting factor for their germination. A light gradient with depth seems to be responsible for the effect of burial on seed germination [52]. Our results provide insight into germination strategies in response to species regeneration and seed bank formation. Strict or conditional primary light requirements for germination, low germination at constant temperatures, and burial inhibition all promote the formation of a persistent seed bank.

The effects of various habitats on seed germination may be due to the fluctuations in temperature and redox potentials [53]. Our results show that substratum had no significant effects on seed germination of *H. dubia*. This result is consistent with previous studies on seeds of *V. natans* [14]. However, the mud substratum increased the seed germination of *O. alismoides* [15].The germination temperature in our substratum experiment was constant. Additionally, oxygen was not the limited factor for *H. dubia* seed germination. When undertaking restoration, nutrient conditions should be taken into consideration. This result indicates that *H. dubia* seeds can germinate under various habitats and nutrients might not be the limiting factor for seed germination.

Previous studies showed negative or no response to high seed densities in some species [54]. In our study, high seed clustering density increased the seed germination of *H. dubia*. This is consistent with previous studies showing that the response of seed germination to seed clustering in Hydrocharitaceae is mostly positive [51]. Similar results were found in seeds of *O. alismoides* [19]. A negative response to high seed densities should reduce the likelihood of intense intraspecific competition. Baskin et al. [55] reported that ethylene can significantly stimulate the seed germination of *Schoenoplectus hallii* (Cyperaceae), a rare summer annual of occasionally flooded sites. Yin et al. [19] supposed that the seeds of *O. alismoides* may produce ethylene, which then stimulates germination. Moreover, positive response to seed clustering density may be related to their reproductive strategy. As a consequence of a gelatinous matter in the fruit of *H. dubia*, the seeds are clustered. The clustering effect ensures that once the clustered seeds land in suitable conditions, most of them will germinate. Germinating seeds of some species release substances that inhibit the germination of other seeds [18]. Thus, they produce a patch area of single species and occupy the new habitat rapidly.

The cardinal diagnostic feature of recalcitrant seeds is that they are sensitive to desiccation and cannot be dried without damage [56,57]. In our study, the moisture content of *H. dubia* seeds under dry storage conditions (4 °C—Dry and Ambient air temperature—Dry) was significantly lower than that in the wet environment (4 °C—Wet and Ambient water temperature—Wet). This indicated that *H. dubia* seeds easily lose water under dry stress. The seed germination of *H. dubia* seeds under dry storage conditions was significantly lower than that in the wet environment. This indicated that *H. dubia* seed viability decreases in a dry environment. Relative humidity affected the moisture content of the seeds, which in turn impacted their vigor [58]. Desiccation-induced changes in biochemical and physiological properties commonly cause the loss of *H. dubia* seed viability [34]. Our results show that the *H. dubia* seed respiration rate under Ambient water temperature—Wet environment was significantly higher than that under Ambient air temperature—Dry environment during seed germination. In ex situ germplasm protection, recalcitrant seeds pose serious challenges, as the seeds are intolerant to desiccation and sensitive to freezing [59]. In our study, the germination of *H. dubia* seed under Ambient water temperature—Wet environment was higher than other storage treatments, staying at 92.8% even after 2 years of storage. Considering the aquatic habitats of *H. dubia*, with its seeds being shed into very moist environments, we conclude that the seeds can be stored under Ambient water temperature—Wet environment to maintain seed viability in 2 years.

## 6. Conclusions and Prospects

The study shows that the germination of *H. dubia* seeds is affected by light, burial depth, and clustering density. Seed germination is significantly promoted by light and high clustering seed density, while burial significantly inhibits it. Oxygen concentration, water depth, and germination substrate do not significantly affect *H. dubia* seed germination.

Regarding the optimal storage method for *H. dubia* seeds, our findings indicate that seeds can maintain high viability when stored in environmental conditions similar to those found in seed banks. In this method of storage, the seeds are submerged at the bottom of a sand-filled container at ambient temperatures ranging from 4 °C to 25 °C.

Our study has shown that *H. dubia* seed germination is significantly influenced by light, burial depth, and clustering density. Therefore, it is important to investigate the mechanism of seed germination by examining the relationship between photochrome and seed germination, as well as the substances that promote seed germination under high clustering density conditions in the future. Additionally, we observed the highest germination and respiration rates in *H. dubia* seeds under ambient water temperature wet storage conditions. Therefore, future studies could explore the maintenance mechanism of seed viability in *H. dubia* by examining changes in enzyme activity during respiration.

## Figures and Tables

**Figure 1 biology-13-00246-f001:**
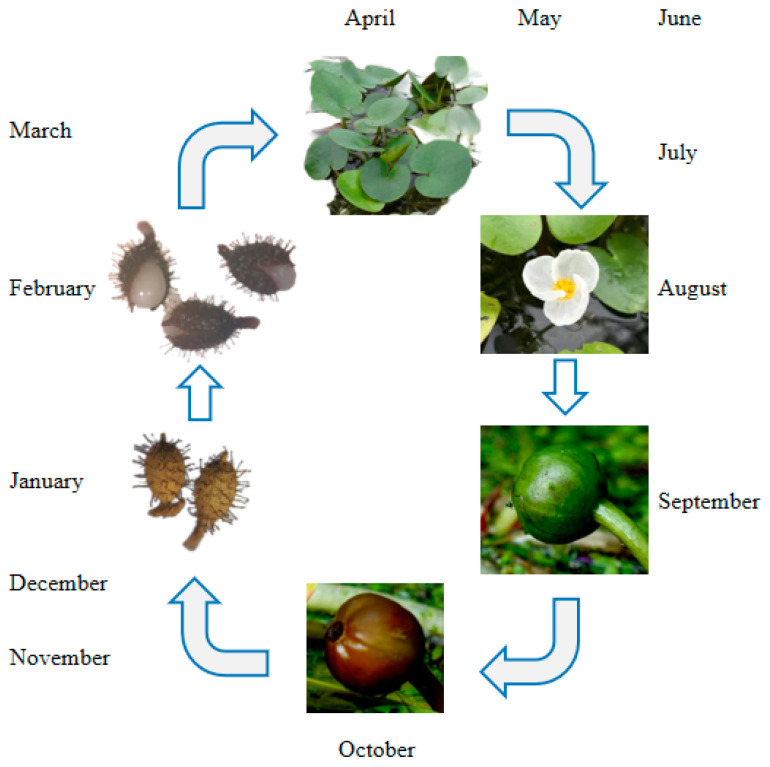
Germination cycle of *Hydrocharis dubia* plant with timeline.

**Figure 2 biology-13-00246-f002:**
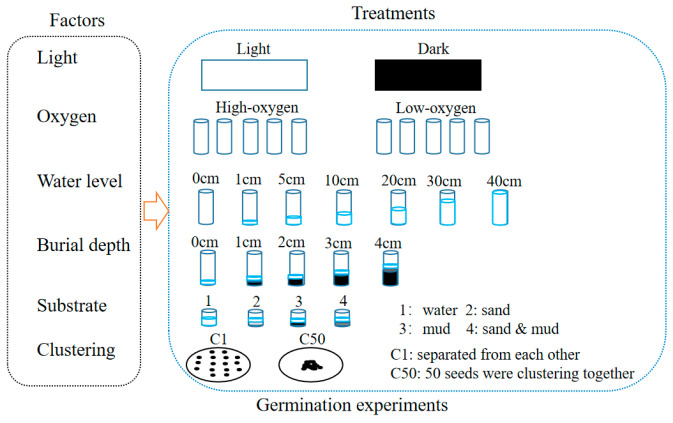
Schematic representation of seed germination experiment design.

**Figure 3 biology-13-00246-f003:**
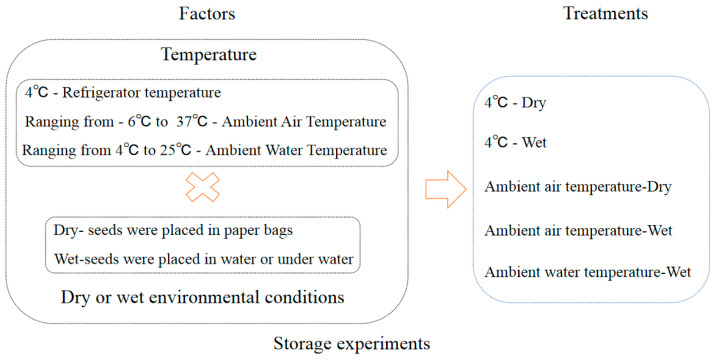
Schematic representation of seed storage experiment design. Note: The initial storage relative humidity of the seeds was 100%. × means “plus”.

**Figure 4 biology-13-00246-f004:**
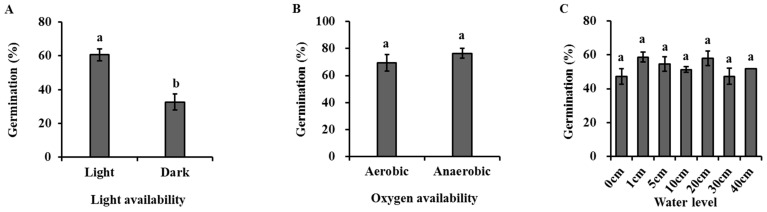
Effects of light (**A**), oxygen availability (**B**), water level (**C**), burial depth (**D**), substratum (**E**), and clustering density (**F**) on germination of *Hydrocharis dubia* seeds. Data represent the mean ± SE (*n* = 5). Data with different letters are significantly different (*p* = 0.05).

**Table 1 biology-13-00246-t001:** Seed storage treatments.

Abbreviation	Temperature	Relative Humidity	Storage Media	Seed Status	Storage Place	Storage Time
4 °C-Dry	4 °C	55%	paper bag	dry	refrigerator	2 years
4 °C-Wet	4 °C	100%	water	wet	refrigerator	2 years
AA-Dry	ambient air temperature ^a^	75%	paper bag	dry	laboratory room	2 years
AA-Wet	ambient air temperature	100%	water	wet	laboratory room	2 years
AW-Wet	ambient water temperature ^b^	100%	wet sand	wet	pond	2 years

^a^ Ambient air temperature range was between −6 °C and 37 °C. Figure A1 provides detailed temperature information. ^b^ Ambient water temperature was between 4 °C and 25 °C.

**Table 2 biology-13-00246-t002:** Analysis of variance for effects of light, oxygen, water level, burial depth, and clustering density on germination in seeds of *Hydrocharis dubia*.

Source	*t*-Value ^a^ or *F*-Value ^b^	*p*
Light availability	4.696 **	0.009
Oxygen availability	−0.979	0.366
Water level	1.601	0.219
Burial depth	69.102 **	<0.001
Substratum	0.984	0.428
Clustering density	−25.298 **	<0.001

^a^ *t*-value in *t*-test. ^b^ *F*-value in one-way ANOVA. ** Significant *p* < 0.01.

**Table 3 biology-13-00246-t003:** Germination, respiration rate and moisture content in seeds of *Hydrocharis dubia* for 2 years under different storage conditions.

Storage Conditions	Germination (%)	Moisture Content (% FW)	Respiration Rate (μmol O_2_ h^−1^ mg^−1^ FW)
4 °C-Dry	64.0 ± 1.7 ^c^	13.80 ± 0.62 ^b^	13.55 ± 1.36 ^ab^
4 °C-Wet	85.2 ± 5.1 ^b^	35.44 ± 0.50 ^a^	11.09 ± 0.51 ^ab^
AA-Dry	0 ± 0 ^d^	11.51 ± 0.57 ^b^	7.36 ± 2.25 ^b^
AA-Wet	-	-	-
AW-Wet	92.8 ± 4.3 ^a^	32.49 ± 1.13 ^a^	15.25 ± 2.10 ^a^

Data represent the mean ± SE (*n* = 5). Data with different letters are significantly different (*p* = 0.05). Seeds in Ambient air temperature—Wet storage environment rotted before the germination experiment.

**Table 4 biology-13-00246-t004:** Analysis of variance for germination, respiration rate, and moisture content in seeds of *Hydrocharis dubia*.

Parameters	*F*-Value ^a^	*p*
Germination	263.499 **	<0.001
Moisture content	201.162 **	<0.001
Respiration rate	11.85 **	<0.001

^a^ *F*-value in one-way ANOVA. ** Significant *p* < 0.01.

## Data Availability

Data are contained within the article.

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
