# Peer review of "Germination Requirement and Suitable Storage Method of Hydrocharis dubia Seeds"

_biology, 2024, doi:10.3390/biology13040246_

Round 1

Reviewer 1 Report (New Reviewer)

Comments and Suggestions for Authors

Hydrocharis dubia  is widely used as an ornamental plant, and therefore its invasive potential is very high. In some regions, these species have already become naturalized, causing significant damage to local ecosystems (Catling et al., 2003; Bean, 2011; Zhu et al., 2018). The conservation status of species of the genus Hydrocharis is assessed as not causing concern, although in regions with active economic activity the area of habitats has significantly decreased, and some populations have declined or disappeared (Efremov et al., 2020). Therefore, in places where they are declining, it is necessary to carry out work to protect and restore them, since Hydrocharis dubia , like other aquatic plants, is necessary as a food supply for many species of aquatic animals and to improve water quality by reducing the availability of nutrients. To successfully preserve and restore plants, it is necessary to study methods for storing seeds and their germination. In connection with the above, the relevance of the research is beyond doubt. Various factors affecting the germination of Hydrocharis dubia  seeds were studied: light, oxygen availability, water level, seeding depth, substrate composition, clustering density and seed storage experiments. All germination experiments were conducted at 122°C using a one-factor design with 5 replicates and 50 seeds per replicate.

 A large experimental work has been carried out. The article is well illustrated; 54 literary sources were used in writing.

The conclusions come from the research results. The authors found that the germination of H. dubia seeds is significantly influenced by light, seed placement depth, and clustering density. Therefore, it is important to investigate substances that promote or inhibit seed germination under given environmental conditions. This will allow us to understand the mechanism of seed germination in the future. It was also determined that the highest germination and respiration rates in H. dubia seeds under ambient water temperature wet storage conditions.

The research was conducted from 2013 to 2015, so this begs the question: why was the article submitted for publication so late?

In my opinion, for the purity of the experiments, it is necessary to take at least 200 seeds for research in each repetition.

Author Response

Reviewer 2 Report (New Reviewer)

Comments and Suggestions for Authors

It would be necessary to adjust the introduction and state the research hypotheses clearly. Describe further the gap that motivated this research.

graphics can be better presented.

The discussion lacks a mechanism of action, almost no results are explained but rather compared with what has already been published for other species. The discussion should move on to the mechanisms, why does light affect the germination of this species? Why is oxygen concentration irrelevant? Just as was done in the cluster variable.

The conclusion is in the present and not in the past and it would be necessary to adjust better, the results are repeated a lot.

Author Response

Reviewer 3 Report (New Reviewer)

Comments and Suggestions for Authors

Kindly find attachment file

Author Response

This manuscript is a resubmission of an earlier submission. The following is a list of the peer review reports and author responses from that submission.

Round 1

Reviewer 1 Report

Comments and Suggestions for Authors

Research into the biology of reproduction of aquatic plants is not only an important environmental task, but also a technologically difficult procedure to implement. The authors have carried out a series of experiments which provide convincing evidence that certain factors influence seed germination Hydrocharis dubia.

 Minor comments and recommendations.

1. The authors did not take into account the results of studies on the reproduction biology of other species of the genus Hydrocharis and seed morphology (Serbanescu-Jitariu, 1972; Cook, Lunnd, 1982; Symoens et al., 1984; Ru et al., 2015; Symoens, 2015; Efremov et al., 2021, Efremov et al., 2023, etc.), which largely determine germination of the target species.

 2. «As a floating-leaved aquatic plant, Hydrocharis dubia could be used as food for animals, or as vegetable, green manure or for medicine». It would be useful to clarify for what medicinal purposes this plant is used.

 3. «…Moreover, it has high seed productivity and it can produce a large amount of seeds every year. Each fruit contains about 200-400 seeds. The seeds can be easily collected. This makes seed banking conservation an effective way to protect and re-seeding, a particularly attractive method for restoration. This plant can reproduce through clonal growth and seed germination, however, there is little information about the factors influencing seed germination and the effective storage condition to maintain seed vigor…».

In most part of the range, species of the genus Hydrocharis are regenerated vegetatively, and the proportion of fructiferous plants is not high. How often do plants in China produce fruits? Is there information on the number of fruits/seeds per unit area? The values of 200-400 seeds/fruit significantly exceed those previously known. It is necessary to clarify how and for which group (cultivated, in vivo  plants) this value was obtained?

   4. «…This demonstrated that the effect of water level on H. dubia seed germination had no relationship with oxygen availability and seeds can get enough light to germinate even when they are submerged at 40cm water level…». Probably 4 cm, based on the results of the experiment.

Reviewer 2 Report

Comments and Suggestions for Authors

The paper contains some interesting data, but upon careful review, this research is still in its preliminary stage, the first draft. The experimental results do not match their tables, such as germination %, which is too high (see Table 3), which is in doubt. I suggest the authors reevaluate the data and/or perform more experimentation to verify and confirm results. Furthermore, the absence of a control group in the research substantially compromises the reliability of the conclusions. It is challenging to ascertain whether the reported effects are related to the experimental manipulation in the absence of a comparison group. I feel that the manuscript is inappropriate for publication in its current form. 

If it is constructive, I have some major comments:

  1. The abstract should concisely overview the paper's key points, including the problem, methodology, findings, and implications. It currently lacks specific information. In addition, there is no numerical data. I suggest carefully rewrite the abstract.
  2. Please recheck the paper and match the results with the respective tables. 
  3. Please re-look all the tables. I think decimal points are missing in the data (For all tables). For example, germination in Table 3 is very high. 
  4. Please include a separate conclusion section and the scope of future research in the conclusion section, as it will strengthen knowledge in this field.  
  5. Important sections are missing such as : Authors contribution, Funding, Funding, Institutional Review Board Statement, Informed Consent Statement, Data Availability Statement, Conflicts of Interest. Please carefully prepare according to authors guidelines.  
  6. Referencing and format are also inconsistent. Please follow the author's guidelines. 

Please see attached file for more comments.

Comments on the Quality of English Language

Reviewer 3 Report

Comments and Suggestions for Authors

In the manuscript entitled, “Germination requirement and suitable storage method of Hydrocharis dubia seeds,” authors did experiments to find the long-term storage condition for H. dubai seeds.  Overall, the article has scope, but it needs improvement. These are the comments to improve the quality of the article.

Comments:

1.       In the experiments on light availability, water level…etc, please also provide details of the oxygen/carbon dioxide conditions used, whether they are anaerobic or aerobic. Please mention in the material method section.

2.       In Table 1, relative humidity has been mentioned. Please explain did you provide the humidity, or if it was calculated after storage.

3.       In Table 2, authors have used ‘t/F function’. Please mention in the table what it represents.

4.       In Table 4, ‘F parameter’ is mentioned. Please provide details of what “F” represents here.

5.       It would be nice if the authors provided a schematic representation of the study design, with detailed experimental plan for each method. It would make the manuscript more comprehensible for the readers.

6.       Please make a figure for germination cycle of Hydrocharis dubia plant with timeline.

7.       On many occasions the full name Hydrocharis dubia has used and some occasion H. dubai has used. Please use full name at first instance and then short form for the consistency. Please use italicize name for the plant scientific name.

Comments on the Quality of English Language

Minor editing required.

Round 2

Reviewer 2 Report

Comments and Suggestions for Authors

Manuscript quality is not sufficiently reliable yet that can it published in a high impact journal like Biology. I previously recommended rejecting this article. Apart from my previous concerns the manuscript lack novelty and looks preliminary report.

Author Response

.

Reviewer 3 Report

Comments and Suggestions for Authors

The Author of the manuscript entitled “Germination requirement and suitable storage method of Hydrocharis dubia seeds” has provided answers of the comments asked previously. The manuscript is in good shape and can be accepted in the biology journal. Before accepting I have one minor comment in the figure 1, that please mention time for each stage of the growing period.

Author Response

.
